# Exponentially Convergent Galerkin Method for Numerical Modeling of Lasing in Microcavities with Piercing Holes

**Alexander O. Spiridonov** [1,*], **Anna I. Repina** [2], **Ilya V. Ketov** [3], **Sergey I. Solov'ev** [4] **and Evgenii M. Karchevskii** [3]

1   Laboratory of Computational Technologies and Computer Modeling, Kazan Federal University, 18 Kremlevskaya St., 420008 Kazan, Russia
2   Department of System Analysis and Information Technologies, Kazan Federal University, 18 Kremlevskaya St., 420008 Kazan, Russia; airepinas@gmail.com
3   Department of Applied Mathematics, Kazan Federal University, 18 Kremlevskaya St., 420008 Kazan, Russia; ketov99@mail.ru (I.V.K.); ekarchev70@gmail.com (E.M.K.)
4   Department of Numerical Mathematics, Kazan Federal University, 18 Kremlevskaya St., 420008 Kazan, Russia; Sergei.Solovyev@kpfu.ru
*   Correspondence: aospiridonov@gmail.com

**Abstract:** The paper investigates an algorithm for the numerical solution of a parametric eigenvalue problem for the Helmholtz equation on the plane specially tailored for the accurate mathematical modeling of lasing modes of microring lasers. The original problem is reduced to a nonlinear eigenvalue problem for a system of Muller boundary integral equations. For the numerical solution of the obtained problem, we use a trigonometric Galerkin method, prove its convergence, and derive error estimates in the eigenvalue and eigenfunction approximation. Previous numerical experiments have shown that the method converges exponentially. In the current paper, we prove that if the generalized eigenfunctions are analytic, then the approximate eigenvalues and eigenfunctions exponentially converge to the exact ones as the number of basis functions increases. To demonstrate the practical effectiveness of the algorithm, we find geometrical characteristics of microring lasers that provide a significant increase in the directivity of lasing emission, while maintaining low lasing thresholds.

**Keywords:** nonlinear eigenvalue problem; boundary integral equation; trigonometric Galerkin method; accuracy estimate; microring laser

## 1. Introduction

Various two-dimensional (2D) models of microdisk and microring lasers (see, e.g., [1,2]) can be investigated with the aid of a specific electromagnetic eigenvalue problem adapted to calculate the threshold values of gain, in addition to the emission frequencies, which is called the lasing eigenvalue problem (LEP) [3–7]. For 2D microcavity lasers with uniform gain, LEP was reduced in [8] to a nonlinear eigenvalue problem for the system of the Muller boundary integral equations (BIEs). This system, obtained by Muller in [9], is widely used in the analysis of electromagnetic-wave scattering from 2D and 3D homogeneous dielectric objects with smooth boundaries [10,11]. This is because Muller BIEs are the Fredholm second-kind equations, which guarantee the convergence of their numerical solutions. By the same reasons, the eigenmodes of fully active [6,8] and passive [12] microcavities can be calculated using Muller BIEs. Many authors, as in [12], have used a physical model called the complex-frequency eigenvalue problem (CFEP). It is based on the search for complex-valued natural frequencies of open passive resonators. To be able to build a general theory for both LEP and CFEP models, a generalized model was proposed in [8]. It obtained the following name: generalized complex-frequency eigenvalue problem (GCFEP) [8]. The reason for reducing GCFEP to the Muller BIEs was to get a system of weakly singular

integral equations [13] on the boundary of the microcavity laser. However, there is no full equivalence between GCFEP and the eigenvalue problem for the system of Muller BIEs [14]. Namely, it was proven in [15] that for each eigenfunction of GCFEP there is a corresponding eigenvector of the system of Muller BIEs. Still, the assertion in the opposite direction is not true: there is one more problem that is reduced to the Muller BIEs, called "turned inside out GCFEP" [15]. If GCFEP and the turned inside out GCFEP together have only the trivial solutions, then the system of Muller BIEs has only the trivial solution [15], and the resolvent set of the corresponding operator-valued function is not empty. This result is important for the theoretical investigation of the spectrum of the eigenvalue problem. Using it and the fundamental results of the theory of projection methods for holomorphic Fredholm operator-valued functions [16,17], the convergence of a Nystrom method was proven in [8].

Recently, for numerical simulation of more complicated 2D microcavity lasers, namely, active cavities with piercing holes [18], a modified version of the Muller BIEs, together with a trigonometric Galerkin discretization technique, was proposed [19,20]. Mathematically, this means that there is an additional region (the hole) inside the cavity domain, and hence, an additional boundary in the integral-equation formulation. This makes the theoretical analysis more difficult compared with [8,14], as well as [15], where the problems with one boundary were investigated, as it was done originally by Muller [9]. In [21], the authors generalized results of [15] and clarified the connection between GCFEP and the eigenvalue problem for the system of Muller BIEs in this more complicated situation.

In [19,20], the authors investigated the directivities, spectra, and thresholds of the on-threshold modes of eccentric microring lasers. For such circular microcavity lasers with non-concentric circular air holes, explicit expressions for the matrix elements were obtained in [19,20]. Together with an account of the symmetry, this made the calculations much faster and more stable. Additionally, the analysis of the numerical experiments in [19,20] demonstrated the exponential convergence of the Galerkin method.

The main idea of the present work is to provide, using results of [21], a rigorous proof of convergence of the Galerkin method proposed previously in [19,20] for the numerical modeling of lasers with piercing holes, and to derive the accuracy estimates for the approximate eigenvalues and eigenfunctions. Our consideration, similar to [8], is based on the fundamental results of the theory of holomorphic operator-valued functions. Using the Galerkin method, we build a sequence of finite-dimensional holomorphic operator-valued functions that regularly approximate the original holomorphic Fredholm operator-valued function. This enables us to apply the results of the general theory to the numerical analysis of the proposed method. Particularly, we prove that if the generalized eigenfunctions are $2\pi$-periodic and analytic in a strip of the complex plane about the real axis, then the approximate eigenvalues and eigenfunctions exponentially converge to the exact ones as the number of the basis functions increases (see Section 3 of the paper, Theorem 4, estimate ii). Thus, the numerical results of [19,20], where the exponential convergence of the Galerkin method was observed, now obtain firm mathematical ground, as for the circular boundaries the generalized eigenfunctions are infinitely smooth.

Preliminarily, in Section 2, we follow [18] and briefly recall the main steps of reducing of the original problem to the nonlinear eigenvalue problem for the set of Muller BIEs. Section 3 presents the equations related to the trigonometric Galerkin discretization of the mentioned BIEs.

To demonstrate the practical efficiency of the algorithm, in Section 4, we show that an air hole in a circular active cavity located at a certain place and with a suitable radius can lead to a notable growth in the directivity of the lasing emission together with the preservation of the low thresholds. This agrees with the physical experiments described in [22].

## 2. GCFEP and Nonlinear Eigenvalue Problem for the Set of Muller BIEs

The formulation of GCFEP for 2D microcavity lasers with piercing holes is given in [18]. A generic geometry of the analyzed microcavities is shown in Figure 1. The air hole is domain $\Omega_1$, the main body of the resonator is denoted as $\Omega_2$, and the environment of the resonator is $\Omega_3$. The boundaries $\Gamma_1$ and $\Gamma_2$ separate these regions. We suppose that the boundaries $\Gamma_1$ and $\Gamma_2$ are twice continuously differentiable, and $n_1$ and $n_2$ are the outer normal unit vectors to them, respectively.

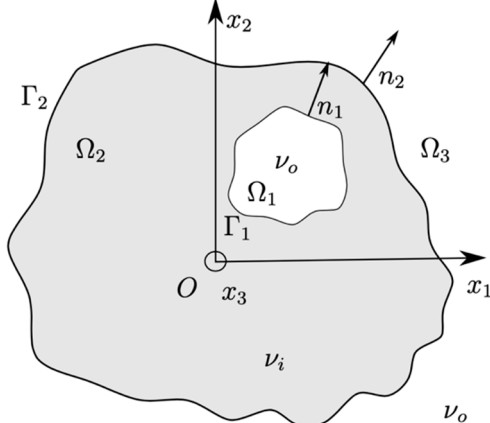

**Figure 1.** Geometry of a 2D microcavity laser with a piercing hole.

Material properties of the laser cavity can be characterized using either the dielectric permittivity or the refractive index. For non-magnetic materials, these two options are equivalent to each other. We used the latter choice because this is customary in optics and photonics research.

Thus, we assumed that the positive refractive index $\nu_o$ of the hole $\Omega_1$ and the environment around the resonator $\Omega_3$ are given. The complex-valued refractive index of the domain $\Omega_2$ is $\nu_i = \alpha_i - i\gamma$. We denote the given real part of $\nu_i$ by $\alpha_i > 0$ and the imaginary part, which is the real-valued parameter of GCFEP, by $\gamma \in \mathbb{R}$. The case of $\gamma = 0$ corresponds to the passive cavity (i.e., without material losses), $\gamma < 0$ is for the cavity with lossy material, and if the region $\Omega_2$ is filled in with a gain material, then $\gamma > 0$. In the latter case, the imaginary part of $\nu_i$ is called the gain index.

We assumed that the electromagnetic field does not depend on the variable $x_3$ and depends on the time, as $\sim \exp(-ikct)$. Herein, the speed of light in a vacuum was denoted by $c$. We were looking for complex values of $k$ on the Riemann surface $\mathbb{L}$ of the function $\ln k$. Because of the independence of the electromagnetic field on the $x_3$ variable, we are dealing with the scalar eigenfunctions of GCFEP $u \in U \backslash \{0\}$, each of which is the third element of the density vector E or H for the E- and H-polarization, respectively. We use the notation $U$ for the space of functions, which are complex-valued and continuous on $\overline{\Omega_1}$, $\overline{\Omega_2}$, and $\overline{\Omega_3}$ and twice continuously differentiable on $\Omega_1$, $\Omega_2$, and $\Omega_3$.

For each $\gamma \in \mathbb{R}$, the eigenvalues $k \in \mathbb{L}$ and the eigenfunctions $u \in U \backslash \{0\}$ of GCFEP have to satisfy the Helmholtz equations,

$$\Delta u + k_o^2 u = 0, \ x \in \Omega_1, \tag{1}$$

$$\Delta u + k_i^2 u = 0, \ x \in \Omega_2, \tag{2}$$

$$\Delta u + k_o^2 u = 0, \ x \in \Omega_3, \tag{3}$$

the transmission conditions,

$$u^- = u^+, \ \eta_o \frac{\partial u^-}{\partial n_1} = \eta_i \frac{\partial u^+}{\partial n_1}, \ x \in \Gamma_1, \tag{4}$$

$$u^- = u^+, \; \eta_i \frac{\partial u^-}{\partial n_2} = \eta_o \frac{\partial u^+}{\partial n_2}, \; x \in \Gamma_2, \tag{5}$$

and the outgoing Reichardt radiation condition [23],

$$u(\rho, \varphi) = \sum_{l=-\infty}^{\infty} a_l H_l^{(1)}(k_o \rho) \exp(il\varphi), \; \rho \geq R_0. \tag{6}$$

Here, the polar coordinates of point $x$ are denoted by $(\rho, \varphi)$, $k_o = kv_o$, $k_i = kv_i$. In Equations (4) and (5), we have the dependence of the coefficients on the polarization; namely, $\eta_{o,i} = v_{o,i}^{-2}$ and $\eta_{o,i} = 1$ for the H- and E-polarization, respectively. The Hankel function of the first kind with the index $l$ is denoted by $H_l^{(1)}(z)$. The functions $u \in U$ in (4) and (5), which are related to the boundary conditions, have the following limit values (see, e.g., [24], p. 68):

$$\frac{\partial u^{\pm}}{\partial n_i}(x) = \lim_{h \to +0} (n_i(x), \operatorname{grad} u(x \pm hn(x)), \; x \in \Gamma_i, \; i = 1, 2, \tag{7}$$

which are expected to exist uniformly on $\Gamma_{1,2}$. The series in (6) converges uniformly and absolutely for any eigenfunction of GCFEP; besides, it is important to note that it is an infinitely term wise differentiable [8].

We denote the main sheet of $\mathbb{L}$ by $\mathbb{L}_0$ and suppose that it is branch-cut along the negative imaginary semi-axis. At this point, we note that three types of GCFEP eigenfunctions exist, depending on the location of the eigenvalue $k \in \mathbb{L}_0$. Equation (6) is interchangeable to the common Sommerfeld radiation condition in the case of $\operatorname{Im} k = 0$,

$$\left( \frac{\partial}{\partial \rho} - ik_o \right) u = o\left( \frac{1}{\sqrt{\rho}} \right), \; \rho \to \infty. \tag{8}$$

The case of $\operatorname{Im} k > 0$ corresponds to the situation when $u$ exponentially decays as $\rho \to \infty$. The alternative case, $\operatorname{Im} k < 0$ entails the eigenfunction $u$ growing exponentially at infinity. An important note for our consideration is that the following property is true [8,18,23] for any $k \in \mathbb{L}, \gamma \in \mathbb{R}$, and $u$, which satisfies (3) and (6):

$$\int_{\Gamma_R} u^-(y) \frac{\partial G_o(x, y)}{\partial n(y)} dl(y) - \int_{\Gamma_R} G_o(x, y) \frac{\partial u^-(y)}{\partial n(y)} dl(y) = 0. \tag{9}$$

Here, $x \in \Omega_3$, $G_o = (i/4) H_0^{(1)}(k_o|x - y|)$. We denote $\Gamma_R$ as the circle with a big enough radius $R$, which center is located at $x$. This fact helps us explore all the eigenfunction types within the same framework.

We need to remember about the dependence of the imaginary part of $k \in \mathbb{L}_0$ on $\gamma \in \mathbb{R}$ [8]. In the case of the passive cavity, where $\gamma \leq 0$, without losses or with them, the GCFEP statement conforms with the usual statement of CFEP [12]. At this point, $\operatorname{Im} k < 0$ for all the eigenvalues $k \in \mathbb{L}_0$. The alternative case is the active cavity, where $\gamma > 0$, and the imaginary part of $k \in \mathbb{L}_0$ can be equal to or greater than zero. The pair $(k, \gamma)$, where $\gamma$ and $k$ are positive, and the corresponding eigenfunction $u$ satisfy all the conditions of LEP [6]. Particularly, condition (8) holds true.

Following [18], we use the integral representations of the eigenfunctions of the problem (1)–(6) in the domains $\Omega_1, \Omega_2$, and $\Omega_3$, respectively,

$$u(x) = -\int_{\Gamma_1} \frac{\partial G_o(x, y)}{\partial n_1(y)} u^-(y) dl(y) + \int_{\Gamma_1} G_o(x, y) \frac{\partial u^-(y)}{\partial n_1(y)} dl(y), \; x \in \Omega_1, \tag{10}$$

$$u(x) = \int_{\Gamma_1} \frac{\partial G_o(x,y)}{\partial n_1(y)} u^+(y) dl(y) - \int_{\Gamma_1} G_o(x, y) \frac{\partial u^+(y)}{\partial n_1(y)} dl(y) - \\ - \int_{\Gamma_2} \frac{\partial G_i(x,y)}{\partial n_2(y)} u^-(y) dl(y) + \int_{\Gamma_2} G_i(x, y) \frac{\partial u^-(y)}{\partial n_2(y)} dl(y), \; x \in \Omega_2, \tag{11}$$

$$u(x) = \int_{\Gamma_2} \frac{\partial G_o(x,y)}{\partial n_2(y)} u^+(y) \mathrm{d}l(y) - \int_{\Gamma_2} G_o(x,y) \frac{\partial u^+(y)}{\partial n_2(y)} \mathrm{d}l(y), \ x \in \Omega_3, \tag{12}$$

where $G_i = (i/4) H_0^{(1)}(k_i |x - y|)$. Equations (10) and (11) are well known (see, e.g., [24], p. 68). Equation (12) also holds true as we have Equation (9) for each value of parameters $k \in \mathbb{L}$ and $\gamma \in \mathbb{R}$ (see [8,18]). Now, we introduce the notations,

$$u_j(x) = u^+(x) = u^-(x), \ x \in \Gamma_j, \ j = 1, 2, \tag{13}$$

$$v_1 = \frac{\eta_i + \eta_o}{2\eta_o} \frac{\partial u^+}{\partial n_1} = \frac{\eta_i + \eta_o}{2\eta_i} \frac{\partial u^-}{\partial n_1}, \ x \in \Gamma_1, \tag{14}$$

$$v_2 = \frac{\eta_i + \eta_o}{2\eta_i} \frac{\partial u^+}{\partial n_2} = \frac{\eta_i + \eta_o}{2\eta_o} \frac{\partial u^-}{\partial n_2}, \ x \in \Gamma_2, \tag{15}$$

and denote the space of continuous on $\Gamma_j$, $j = 1, 2$, functions with the maximum norm by $C_j = C(\Gamma_j)$, $j = 1, 2$, $C = C_1 \times C_2$, and $W = C \times C$. Furthermore, we denote the identical operator in the space $W$ by $I$. Then, any solution of GCFEP (1)–(6) in terms (13)–(15) satisfies the following nonlinear eigenvalue problem for the set of Muller BIEs [18]:

$$A(k, \gamma)w = (I + B(k, \gamma))w = 0, \tag{16}$$

$$B = \begin{pmatrix} B_1^{(1,1)} & B_1^{(1,2)} & B_1^{(1,3)} & B_1^{(1,4)} \\ B_1^{(2,1)} & B_1^{(2,2)} & B_1^{(2,3)} & B_1^{(2,4)} \\ B_2^{(3,1)} & B_2^{(3,2)} & B_2^{(3,3)} & B_2^{(3,4)} \\ B_2^{(4,1)} & B_2^{(4,2)} & B_2^{(4,3)} & B_2^{(4,4)} \end{pmatrix}, \ w = \begin{pmatrix} u_1 \\ v_1 \\ u_2 \\ v_2 \end{pmatrix}$$

$$\left( B_j^{l,m}(k, \gamma)g \right)(x) = \int_{\Gamma_j} K_j^{(l,m)}(k, \gamma; x, y) g(y) \mathrm{d}l(y).$$

Here, we denote $u_j$ or $v_j$, $j = 1, 2$ by the function $g$. The kernels have the following forms [18]:

$$K_j^{(1,1)} = -K_j^{(3,3)} = \frac{\partial G_o(x,y) - \partial G_i(x,y)}{\partial n_j(y)}, \ x \in \Gamma_j, \ y \in \Gamma_j, \ j = 1, 2,$$

$$K_j^{(1,2)} = -K_j^{(3,4)} = \frac{2(\eta_o G_i(x,y) - \eta_i G_o(x,y))}{\eta_i + \eta_o}, \ x \in \Gamma_j, \ y \in \Gamma_j, \ j = 1, 2,$$

$$K_1^{(1,3)} = \frac{\partial G_i(x,y)}{\partial n_2(y)}, \ K_1^{(1,4)} = -\frac{2\eta_o \partial G_i(x,y)}{\eta_o + \eta_i}, \ x \in \Gamma_1, \ y \in \Gamma_2,$$

$$K_j^{(2,1)} = -K_j^{(4,3)} = \frac{\partial^2 G_o(x,y)}{\partial n_j(x)\partial n_j(y)} - \frac{\partial^2 G_i(x,y)}{\partial n_j(x)\partial n_j(y)}, \ x \in \Gamma_j, \ y \in \Gamma_j, \ j = 1, 2,$$

$$K_j^{(2,2)} = -K_j^{(4,4)} = \frac{2\eta_o}{\eta_o + \eta_i} \frac{\partial G_i(x,y)}{\partial n_j(y)} - \frac{2\eta_i}{\eta_o + \eta_i} \frac{\partial G_o(x,y)}{\partial n_j(y)}, \ x \in \Gamma_j, \ y \in \Gamma_j, \ j = 1, 2,$$

$$K_1^{(2,3)} = \frac{\partial^2 G_i(x,y)}{\partial n_1(x)\partial n_2(y)}, \ K_1^{(2,4)} = -\frac{2\eta_o}{\eta_o + \eta_i} \frac{\partial G_i(x,y)}{\partial n_1(y)}, \ x \in \Gamma_1, \ y \in \Gamma_2,$$

$$K_2^{(3,1)} = -\frac{\partial G_i(x,y)}{\partial n_1(y)}, \ K_2^{(3,2)} = \frac{2\eta_o G_i(x,y)}{\eta_o + \eta_i}, \ x \in \Gamma_2, \ y \in \Gamma_1,$$

$$K_2^{(4,1)} = -\frac{\partial^2 G_i(x,y)}{\partial n_1(y)\partial n_2(x)}, \ K_2^{(4,2)} = -\frac{2\eta_o}{\eta_o + \eta_i} \frac{\partial G_i(x,y)}{\partial n_2(y)}, \ x \in \Gamma_2, \ y \in \Gamma_1.$$

Some of the kernels $K_j^{(q,s)}$ have logarithmic singularities and the others are continuous [13]. Consequently, the operator $B(k, \gamma) : W \to W$ is compact, and the operator $A(k, \gamma) : W \to W$ is Fredholm with index zero for every $k \in \mathbb{L}$ and $\gamma \in \mathbb{R}$ [13].

If $u \in U$ is an eigenfunction of problem (1)–(6) corresponding to an eigenvalue $k \in \mathbb{L}$ for a value of the parameter $\gamma \in \mathbb{R}$, then, defined by (13)–(15), functions $u_j$ and $v_j$ belong to the Banach spaces $C_j$, $j = 1, 2$, respectively, and form a nontrivial solution $w \in W$ of (16) with the same values of $k$ and $\gamma$. This was proved in Theorem 3 of [21]. The assertion in the opposite direction relative to the statement of this theorem is not true, as, as in [18], we did not substitute representations (10)–(12) into (4) and (5), but added the limit values of them and their normal derivatives from both sides of the boundaries $\Gamma_1$ and $\Gamma_2$ term by term. However, the following result holds true (see Theorem 4 [21]). For each $\gamma \in \mathbb{R}$ and $k \in \mathbb{I}_+$ problem (16) has only the trivial solution $w = 0$, $w \in W$. Here, $\mathbb{I}_+$ denotes the strictly positive imaginary semi-axis of $\mathbb{L}_0$.

## 3. Galerkin Method

In the current section, we present a trigonometric Galerkin method for the numerical solution of problem (16). Assume that each contour $\Gamma_j$ has a parameterization $\rho_j(t) = \left(\rho_j^1(t), \rho_j^2(t)\right)$, where $\rho_j^1(t) = f_j(t)\cos t$, $\rho_j^2(t) = f_j(t)\sin t$, $t \in [0, 2\pi]$, $j = 1, 2$. Then, for any given $\gamma \in \mathbb{R}$, we have

$$\left(B^{(l,m)}(k)w^{(m)}\right)(t) = \frac{1}{2\pi}\int_0^{2\pi} K^{(l,m)}(k; t, \tau)w^{(m)}(\tau)\mathrm{d}\tau.$$

Here, $l$, $m = 1, 2, 3, 4$, $y = y(\tau) \in \Gamma_j$, $j = 1, 2$,

$$K^{(l,m)}(k; t, \tau) = 2\pi K_j^{(l,m)}(k; x, y), \ w^{(m)}(\tau) = w^{(m)}(y)\left|\rho_j'(\tau)\right|.$$

For the construction and investigation of the Galerkin method, it is convenient to consider the problem (16) in the Hilbert space $H = (L_2)^4$, where $L_2$ denotes the space of square integrable functions with the inner product

$$(u, v) = \frac{1}{2\pi}\int_0^{2\pi} u(\tau)\overline{v(\tau)}\mathrm{d}\tau, \ u, v \in L_2.$$

By $T_n \subset L_2$, we denote the subspace of all trigonometric polynomials of the order no greater than $n$ with complex coefficients. Then, $H_n = (T_n)^4 \subset H$ is the subspace with the elements of the form,

$$\mathrm{w}_n = \begin{pmatrix} w_n^{(1)} \\ w_n^{(2)} \\ w_n^{(3)} \\ w_n^{(4)} \end{pmatrix}, \ w_n^{(1)}, w_n^{(2)}, w_n^{(3)}, w_n^{(4)} \in T_n.$$

By $p_n : H \to H_n$, we define the following projection operator:

$$p_n w = \begin{pmatrix} \Phi_n w^{(1)} \\ \Phi_n w^{(2)} \\ \Phi_n w^{(3)} \\ \Phi_n w^{(4)} \end{pmatrix}, \ w^{(1)}, w^{(2)}, w^{(3)}, w^{(4)} \in L_2.$$

Here, $\Phi_n : L_2 \to T_n$ is the Fourier operator,

$$\left(\Phi_n w^{(m)}\right)(t) = \sum_{q=-n}^{n} c_q\left(w^{(m)}\right)\varphi_q(t), \ m = 1, 2, 3, 4.$$

For $q = -n, \ldots, n$, the vectors $\varphi_q(t) = \exp(iqt)$ form the orthonormal basis in the space $T_n$. We rewrite Equation (16) as follows

$$w^{(l)} + \sum_{m=1}^{4} B^{(l,m)}(k)w^{(m)} = 0, l = 1, 2, 3, 4. \tag{17}$$

We look for approximate solutions $w_n^{(1)}, w_n^{(2)}, w_n^{(3)}, w_n^{(4)} \in T_n$ of the system of Equation (17) in the form

$$w_n^{(m)}(t) = \sum_{q=-n}^{n} \alpha_q^{(m)} \varphi_q(t), \ n \in N, \ m = 1, 2, 3, 4.$$

Therefore, we have

$$w_n^{(l)} + \sum_{m=1}^{4} B^{(l,m)}(k)w_n^{(m)} = 0, \ l = 1, 2, 3, 4, \ n \in N.$$

We calculate the unknowns $\alpha_q^{(m)}$ using the Galerkin method,

$$\left( w_n^{(l)}, \varphi_p \right) + \sum_{m=1}^{4} (B^{(l,m)}(k)w_n^{(m)}, \varphi_p) = 0, \ p = -n, \ldots, n, \tag{18}$$

where $l = 1, 2, 3, 4$. As the trigonometric functions are orthonormal, we can rewrite Equation (18) in the form of the following system of linear algebraic equations:

$$\alpha_p^{(l)} + \sum_{m=1}^{4} \sum_{q=-n}^{n} h_{pq}^{(l,m)}(k)\alpha_q^{(m)} = 0, \ p = -n, \ldots, n, \tag{19}$$

where $l = 1, 2, 3, 4$,

$$h_{pq}^{(l,m)}(k) = \frac{1}{4\pi^2} \int_0^{2\pi} \int_0^{2\pi} K^{(l,m)}(k; t, \tau) \exp(-ipt) \exp(iq\tau) dt d\tau.$$

The system of linear algebraic Equation (19) is equivalent to the finite-dimensional linear operator equation

$$A_n(k)w_n \equiv p_n A(k)w_n \equiv (I + p_n B(k))w_n \equiv (I + B_n(k))w_n = 0. \tag{20}$$

Here, $k \in L$, $A_n : H_n \to H_n$, $I$ is the unitary operator in the space $H_n$. As usual, we denote by $\rho(A_n)$ and by $\sigma(A_n)$, *the regular and the characteristic sets of the operator-valued function $A_n(k)$*, respectively. Let also $N'$, $N''$, $N'''$, $\ldots$ be infinite sequences of the set of all natural numbers $N$.

**Theorem 1.** *For any given $\gamma \in \mathbb{R}$, the following statements are true:*

1. *If $k_0$ is an eigenvalue of $A(k)$, then for each $n \in N$ there exists an eigenvalue $k_n$ of $A_n(k)$ such that $k_n \to k_0 \ (n \in N)$.*
2. *If for each $n \in N$ there exists an eigenvalue $k_n$ of $A_n(k)$, such that $k_n \to k_0 \in \mathbb{L} \ (n \in N)$, and $w_n$ is a normalized eigenfunction of $A_n(k_n)$, then*
   (i)     *$k_0$ is an eigenvalue of $A(k)$,*
   (ii)    *$\{w_n\}_{n \in N}$ is a discretely compact sequence and its cluster points are normalized eigenfunctions of $A(k_0)$.*
3. *For every compact $L_0 \subset \rho(A)$, the sequence $\{A_n(k)\}_{n \in N}$ is stable on $L_0$, i.e., there exist $n(L_0)$ and $c(L_0)$, such that $L_0 \subset \rho(A_n)$, $A_n(k)^{-1} \leq c(L_0)$ for all $k \in L_0$ and $n \geq n(L_0)$.*

The proof of this theorem is based on the general results of the discrete convergence theory [25] applied for the investigation of approximate methods in the eigenvalue problem, where the parameter appears non-linearly [16]. Therefore, let us preface it with some definitions from [16].

As it is said, the sequence $\{w_n\}_{n \in N'}$ of vectors from the space $H_n$ discretely converges to the limit $w \in H$ if $\|w_n - p_n w\| \to 0$ as $n \to \infty$, $n \in N'$. Discrete convergence of the vectors will be denoted as $w_n \to w(n \in N')$. The sequence of elements $\{w_n\}_{n \in N'}$ is called discretely compact if, for each subsequence $\{w_n\}_{n \in N''}$, $N'' \subseteq N'$, there exists a subset $N''' \subseteq N''$ and a vector $w \in H$, such that $w_n \to w(n \in N''')$.

Consider a bounded linear operator $A \in \mathcal{L}(H, H)$ and a sequence of finite-dimensional bounded linear operators $\{A_n\}_{n \in N}$. It is said that the sequence of operators $\{A_n\}_{n \in N}$ approximate the operator $A$, if for any vector $w \in H$ we have

$$\|A_n p_n w - p_n A w\| \to 0 (n \to \infty)$$

If the discrete convergence of vectors $w_n \to w(n \in N)$ implies the discrete convergence of their images, $A_n w_n \to A w(n \in N)$, then the sequence of operators $\{A_n\}_{n \in N}$ is said to converge discretely to $A$.

The sequence of operators $\{A_n\}_{n \in N}$ is regular if from the boundedness of the sequence of vectors $\{w_n\}_{n \in N}$ (thanks to the estimate $\|w_n\| \leq \mathrm{const}(n \in N)$) and from the discrete compactness of the sequence of their operator images $\{A_n w_n\}_{n \in N}$ follows the discrete compactness of the sequence of the vectors $\{w_n\}_{n \in N}$. If a sequence of operators $\{A_n\}_{n \in N}$ is regular and wherein approximates the operator $A$, then it is said that it regularly approximates $A$. The regular convergence of a sequence of operators is defined in similar way.

It is said that a sequence of operator-valued functions $\{A_n(k)\}_{n \in N}$ regularly converges on $\mathbb{L}$ to an operator-valued function $A(k)$, if for each converging numerical sequence $k_n \to k_0 \in \mathbb{L}(n \to \infty)$, the operator sequence $\{A_n(k_n)\}_{n \in N}$ regularly converges to the operator $A(k_0)$.

**Proof.** Let us verify, that in the case under consideration, all conditions (b1)–(b5) of Theorem 2 of [16] are satisfied. Then, all the assertions of Theorem 1 hold true.

(b1) The operator-valued function $A(k) : H \to H$ is holomorphic and Fredholm on $\mathbb{L}$, and its regular set is not empty. The holomorphicity and the Fredholm property of $A(k) : W \to W$ were proved in Theorem 2 in [18]. For $A(k) : H \to H$, these properties are established similarly with the replacement of estimates for all norms in the space of continuous functions $W$ on the corresponding estimates in the space of functions integrable with the square $H$.

In Theorem 4 from [21], it was established that, for any $k \in \mathbb{I}_+$, Equation (16) has only a trivial solution in the space $W$. The operator $B(k)$ is weakly singular, therefore, any solution of Equation (16) from $H$ must belong to the space $W$ and can only be trivial for $k \in \mathbb{I}_+$. The operator-valued function $A(k) : H \to H$ is a Fredholm one, so for it we have $\mathbb{I}_+ \subset \rho(A)$.

(b2) For any $n \in N$, the operator-valued function $A_n(k) : H_n \to H_n$ is holomorphic and Fredholm on $\mathbb{L}$. Indeed, $A(k) : H \to H$ is holomorphic on $\mathbb{L}$. As the operator $p_n$ is linear and bounded, $A_n(k) = p_n A(k) : H_n \to H_n$ has the same property. The Fredholm property of the operator-valued function $A_n(k)$ is obvious because of its finite-dimensionality.

(b3) On each compact set $L_0 \subset \mathbb{L}$, the norms $\|A_n(k)\|$ are bounded uniformly in the parameters $n \in N$ and $k \in \mathbb{L}_0$. Indeed, from the definition of the operator $A_n(k)$ and the equality

$$\|p_n\| = 1, \tag{21}$$

It follows that

$$\|A_n(k)\| \leq \|A(k)\|, \ n \in N, \ k \in \mathbb{L},$$

However, because of

$$\|B^{(l,m)}(k)\|^2 \le \frac{1}{4\pi^2} \int_0^{2\pi} \int_0^{2\pi} \left|K^{(l,m)}(k;t,\tau)\right|^2 \mathrm{d}t\mathrm{d}\tau, \; l, \; m = 1,2,3,4,$$

The following estimate is correct:

$$\|A(k)\| \le c(k), \; k \in \mathbb{L}, \tag{22}$$

where $c(k)$ is a continuous function on $\mathbb{L}$:

$$c(k) = 1 + \frac{1}{2\pi} \sum_{l,m=1}^{4} \left( \int_0^{2\pi} \int_0^{2\pi} \left|K^{(l,m)}(k;t,\tau)\right|^2 \mathrm{d}t\mathrm{d}\tau \right)^{1/2}.$$

It is easy to see that to complete the verification of the required property, it suffices to compute the maximum of the function $c(k)$ on the given compact set $L_0 \subset \mathbb{L}$.

(b4) For each fixed value $k \in \mathbb{L}$, the operator sequence $\{A_n(k)\}_{n \in N}$ approximates the operator $A(k)$. Indeed, by the definition of the operators $A(k) : H \to H$ and $p_n : H \to H_n$, for any vector $w \in H$ we have

$$\|A_n(k)p_n w - p_n A(k)w\| = \|p_n A(k)p_n w - p_n A(k)w\| \le \|p_n\| \|A(k)\| \|p_n w - w\| \to 0 (n \in N).$$

The tendency to zero is a consequence of the tendency to zero of the norm of the remainder term of the segment of the Fourier series for any function from $L_2$, estimate (22), and equality (21).

(b5) For each fixed value $k \in \mathbb{L}$, the operator sequence $\{A_n(k)\}_{n \in N}$ is regular. Indeed, the discrete compactness of the sequence of vectors $\{A_n(k)w_n\}_{n \in N}$ means that for any $N' \subseteq N$, there exist $N'' \subseteq N\prime$, such that the sequence $\{A_n(k)w_n = w_n + B_n(k)w_n\}$, $n \in N''$ converges discretely to some $z \in H$. If the sequence $\{w_n\}_{n \in N''}$ is bounded, then there is a weakly convergent subsequence $\{w_n\}_{n \in N'''}$, $N''' \subset N''$. As it is known, the compact operator $B(k)$, takes it to a strongly converging one to some vector $u \in H$:

$$\|B(k)w_n - u\| \to 0, \; n \in N'''$$

Hence, by virtue of the inequality

$$\|B_n(k)w_n - p_n u\| \le \|p_n\| \|B(k)w_n - u\|$$

and equality (21), it follows that the sequence $\{B_n w_n\}_{n \in N'''}$ converges discretely to $u \in H$. Thus, $\{w_n\}_{n \in N'''}$ converges discretely to the vector $w = z - u \in H$, and the definition of the regularity of the sequence $\{A_n(k)\}_{n \in N}$ is satisfied. □

As usual, we denote various positive constants that do not depend on $n$ by the same letter $c$. Let $k_0$ be an eigenvalue of $A(k)$. We denote by $G(A, k_0)$ the generalized eigenspace, i.e., the closed linear hull of all the generalized eigenfunctions of $A(k)$ corresponding to $k_0$. As the operator $p_n$ is linear, the next theorem follows from [17].

**Theorem 2.** *Assume that $\gamma \in \mathbb{R}$ is given, $k_0$ is an eigenvalue of $A(k)$, and $L_0 \subset \mathbb{L}$ is a compact set with the boundary $\Gamma_0 \subset \rho(A)$ so that $L_0 \cap \sigma(A) = \{k_0\}$. For each $n \in N$, we denote by $\varepsilon_n$ the maximum of the approximation error over $k \in \Gamma_0$ and $w \in G(A, k_0)$,*

$$\varepsilon_n = \sup\|\{A_n(k)p_n w - p_n A(k)w\| : \; k \in \Gamma_0, \; w \in G(A, k_0), \; \|w\| = 1\}. \tag{23}$$

*Then, $\varepsilon_n \to 0 \; (n \in N)$ and the following estimations hold for almost all $n \in N$:*

(i)　$|k_n - k_0| \le c\varepsilon_n^{1/\kappa}$ *for all $k_n \in \sigma(A_n) \cap L_0$, where $\kappa = \kappa(k_0, A)$ is the multiplicity of the pole $k_0$ of the operator-valued function $A^{-1}(k)$;*

*(ii)* $\left|\bar{k}_n - k_0\right| \leq c\varepsilon_n$, *where $\bar{k}_n$ is the weighted (proportionally to their algebraic multiplicities) mean of all the eigenvalues of $A_n(k)$ in $L_0$, $\bar{k}_n = \sum_{k\in\sigma(A_n)\cap L_0}\mu_k \cdot k$, $\mu_k = \nu(k, A_n)/\nu(k, A)$, where $\nu(\cdot, \cdot)$ is the algebraic multiplicity of the corresponding eigenvalue $k$;*

*(iii)* $max\{|k_n - k_0| : k_n \in \sigma(A_n) \cap L_0\} \leq c\varepsilon_n^{1/l_n}$, *where $l_n$ is the number of the different eigenvalues of $A_n$ in $L_0$.*

The next theorem follows from [26].

**Theorem 3.** *Suppose that the conditions of Theorem 2 are fulfilled, $\varepsilon_n$ is defined in (23), $G_0(A, k_0)$ is the eigenspace of $A(k)$ corresponding to the eigenvalue $k_0 \in \mathbb{L}$, $\{k_n\}_{n\in N}$ and $\{w_n\}_{n\in N}$ are some sequences of eigenvalues $k_n$ of $A_n(k)$ and normalized eigenfunctions $w_n$ of $A_n(k)$, such that $k_n \to k_0$ $(n \in N)$, and $\delta_n$ is defined by the equality*

$$\delta_n = max\|\{A_n(k_0)p_n w_0 - p_n A(k_0)w_0\| : w_0 \in G_0(A, k_0), \|w_0\| = 1\}. \qquad (24)$$

*Then, for each eigenfunction $w_n$ there exists an eigenfunction $w_0 = w_0(w_n) \in G_0(A, k_0)$, such that the following error estimate holds for almost all $n \in N$:*

$$\|w_n - w_0\| \leq c\left(\varepsilon_n^{1/\kappa} + \delta_n\right).$$

Using the results from [27,28], pp. 270, 271, we derive the following approximation error estimates.

**Theorem 4.** Suppose that the conditions of Theorem 2 are fulfilled, $\varepsilon_n$ is defined in (23), $\delta_n$ is defined in (24), and $G_{\eta,m}(0, 2\pi)$ is the Gevrey space [28], p. 271. Then, the following error estimates are valid:

*(i)* $max\{\varepsilon_n, \delta_n\} \leq cn^{-m}$, *when $d^i w/dx^i \in L_2$, $i = 0, 1, \ldots, m$, for any generalized eigenfunction $w \in G(A, k_0)$;*

*(ii)* $max\{\varepsilon_n, \delta_n\} \leq cn^{-m}e^{-\eta n}$, *when $w \in G_{\eta,m}(0, 2\pi)$, $\eta > 0$, $m \geq 0$, for any generalized eigenfunction $w \in G(A, k_0)$.*

We solved the nonlinear eigenvalue problem (20) using the residual inverse iteration algorithm [29]. If the boundaries of the active cavity and the piercing hole were nonconcentric circles, then the entries of the Galerkin's matrix had the explicit expressions calculated carefully in [19,20]. We used them in the next section.

## 4. Numerical Results

Optimization of geometrical parameters of the microring resonator is aimed at finding such microlaser configurations for which a high value of the directivity $D$ (see exact definition in [19,20]) and a low value of the threshold $\gamma$ will be obtained. The modes that meet these requirements are of primary interest during the design of microlasers.

The studies were carried out for the H-polarization, because, for thin flat 3D laser cavities, which can be approximated with 2D models, the values of the thresholds for the H-polarized modes are lower than of the E-polarized modes [5]. This is because such a reduction dimensionality entails the replacement of the bulk refractive index with its effective value, which depends on the polarization. The results were obtained, as in [19,20], for the following parameter values: the refractive index in the domain $\Omega_3$ and in the hole is equal $\nu_e = 1$, the real part of the refractive index in the active region is $\alpha_i = 2.63$, the dimensionless quantities are $\kappa = ka_2$, $d = |O_1 - O_2|/a_2$, $r = a_1/a_2$. Here, the domains $\Omega_1$ and $\Omega_2$ are circles with centers $O_1$ and $O_2$ and radii $a_1$ and $a_2$, respectively.

In our analysis, we used the same mode classification as in [19,20], with index $e$ and $o$ denoting the even and odd eigenfunction symmetry, respectively, with respect to the line of symmetry, which is the $x_1$-axis. We note that, in the ideally circular cavity, the modes with

small radial indices had their fields compressed to the cavity rim. This feature is reflected by their specific name as whispering-gallery modes.

Let us first investigate the dependence of the directivity $D$ on the relative distance $d$ between the center of the cavity and the center of the hole, and the relative radius of the hole $r$ for the modes (11, 1, e/o). Figure 2 shows the dependence of $D$ on $d$ and $r$ in the region, where the cavity contours do not cross each other. The value of $D$ practically does not increase if the values of the parameters $(r, d)$ satisfy the inequality $d + 0.8r \leq 0.4$. Above the straight line described by the equation $d = -0.8r + 0.4$, an increase in the directivity coefficient $D$ was observed for both modes. The regions of values $(r, d)$ in which $D$ is maximal are clearly distinguishable. For the odd mode, this is a vicinity of the point 0.32 and 0.49, for the even mode, this is a vicinity of the point 0.02 and 0.63. We see that the value of $D$ for the even modes is higher than for the odd modes, in addition, obtaining quasi-unidirectional emission is impossible for odd modes [19,20]. Therefore, we carried out further studies for the mode (11, 1, e).

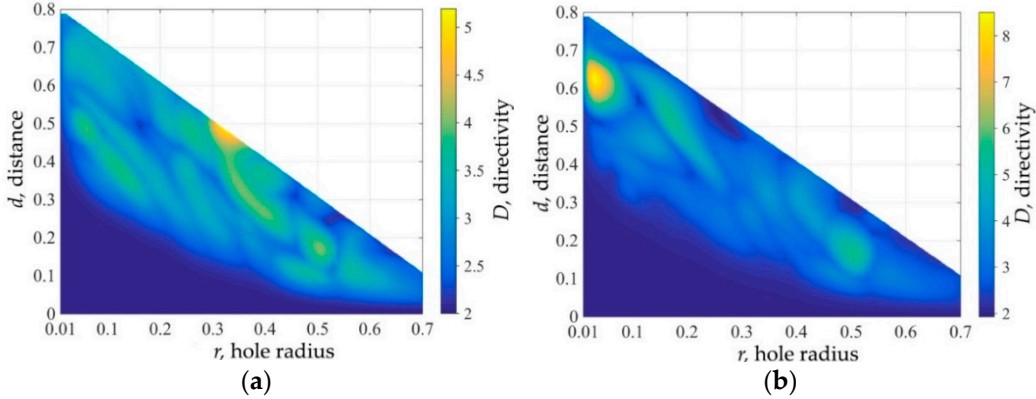

**Figure 2.** Dependence of the directivity $D$ for the modes (11, 1, e/o) on the relative distance $d$ between the centers of the cavity and the hole and the relative radius $r$ of the hole, (**a**) odd mode, and (**b**) even mode.

In addition to obtaining a high value of for directivity $D$, it was necessary to maintain low values of the threshold $\gamma$. It is more convenient to search for low values of $\gamma$ by maximizing the values of the function $T = -\log_{10} \gamma$. Figures 3 and 4 show the dependences of the normalized wavenumber $\kappa$ and the threshold gain index $\gamma$ (in short, the threshold) for the mode (11, 1, e) on the relative distance $d$ between the centers of the cavity and the hole and the relative radius of the hole $r$. We see that the normalized wavenumber $\kappa$ ranges from 5.85 to 6.3. Mostly, $\kappa$ takes on values close to 5.85 and increases only when the cavity contours are close to each other. The values of $\gamma$ remain low in the region under the straight line $d = -0.7r + 0.5$ and in a small rectangle with vertices A = (0.01, 0.5), B = (0.01, 0.8), C = (0.02, 0.5), and D = (0.02, 0.8). Furthermore, above the straight line $d = -0.7r + 0.5$, the values of $\gamma$ increase.

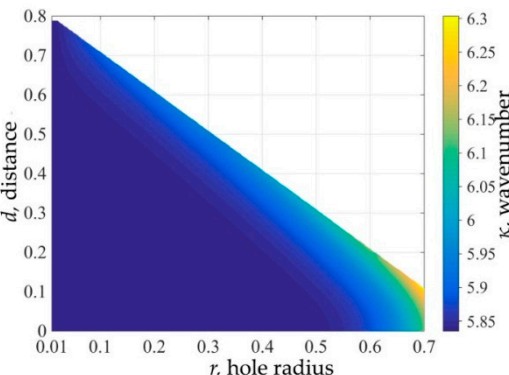

**Figure 3.** Dependence of the normalized wavenumber $\kappa$ for the mode (11, 1, e) on the relative distance $d$ between the centers of the cavity and the hole and the relative radius $r$ of the hole.

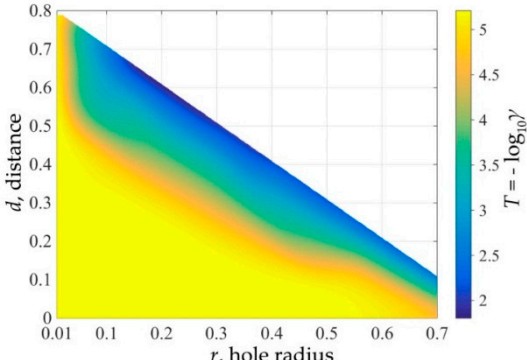

**Figure 4.** Dependence of the threshold $\gamma$ for the mode (11, 1, e) on the relative distance $d$ between the centers of the cavity and the hole and the relative radius $r$ of the hole.

We searched for such pairs of values of $d$ and $r$, for which $D$ took a high value, while at the same time the value of $\gamma$ remained low enough. For this, we considered the target functions of the following forms:

$$F = TD, \tag{25}$$

$$F = T + D, \tag{26}$$

$$F = T + 10D, \tag{27}$$

where $T = -\log_{10}\gamma$. It was assumed that among the points of the local maxima of the target functions (25)–(27), we would find such pairs of values $(r, d)$ for which a high value of the directivity $D$ and a low value of the threshold $\gamma$ would be obtained. For the found points of the local maxima, a check should be performed with a control value. The value of the target function must be no less than the value obtained if the problem for the microdisk resonator (without an air hole) is considered. Solving the problem for the microdisk resonator with the same radius $a_2$, and the same refractive index $\alpha_i$, we have $T = 5.2074$, $D = 2$. This means that, for the target function (25), the control value is $F = 10.4147$, for the target function (26), the control value is $F = 7.2074$, and for the target function (27), the control value is $F = 25.2074$.

Figure 5 shows the points of local maxima of the considered target functions. For function (25), there are nine points of the local maxima; for function (26), there are eight points of the local maxima; and for target function (27), there are four points of the local maxima. Observing the results, we see that some points are presented on all three panels of Figure 5. Next, by intersecting the sets of the local maxima of the considered target functions, for further research, we chose the following pairs of values $(r, d)$: (0.02 and 0.62), (0.5 and 0.15), and (0.17 and 0.5). For the function (25) in Figure 5a, the points under

consideration are numbered as 1, 2, and 4, respectively. For functions (26) and (27) in panels (b) and (c) of Figure 5, the points under consideration are 1, 2, and 3, respectively.

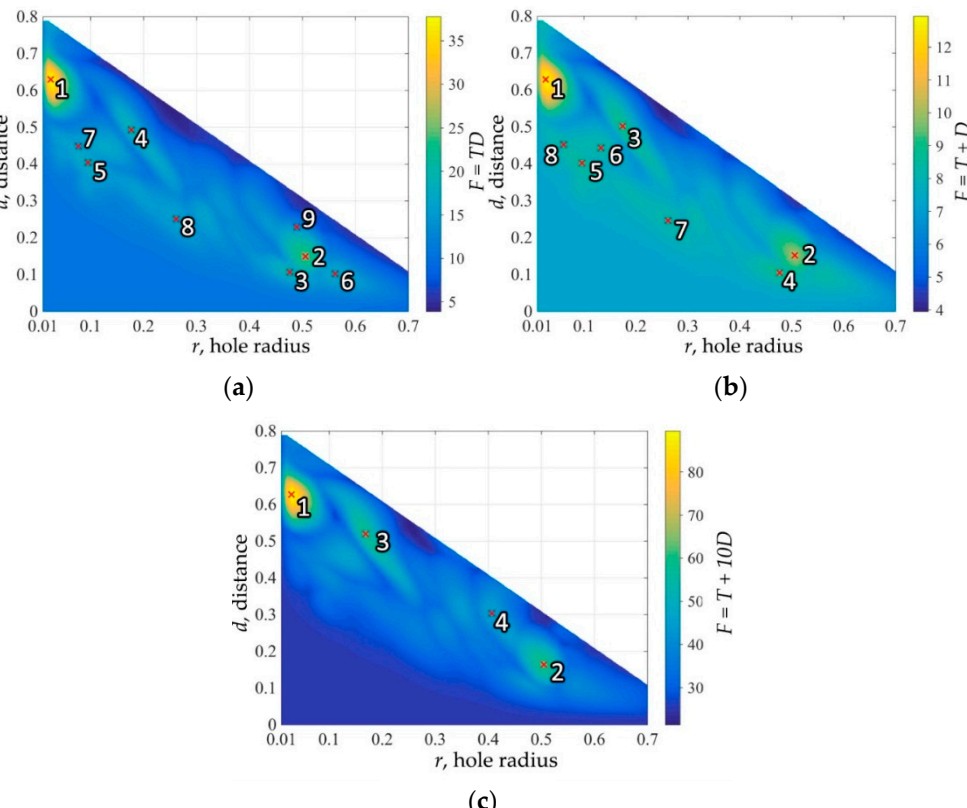

**Figure 5.** Dependence of the values of the target function for the mode (11, 1, e) on the relative distance $d$ between the centers of the cavity and the hole and the relative radius $r$ of the hole, (**a**) $F = TD$, (**b**) $F = T + D$, and (**c**) $F = T + 10D$.

Tables 1–3 show the values of the numerical characteristics of the cavity, and the lasing modes corresponding to the points of the local maximum. Here, $\beta$ is the angle showing the emission direction, i.e., the target of the main beam in the far-field patterns (see Figure 6). The points in the tables are numbered in descending order of the value of the corresponding target function.

**Table 1.** Local maxima of the function $F = TD$.

| No. | $r$ | $d$ | $\beta_{\text{radian}}$ | $\beta_{\text{degrees}}$ | $T$ | $D$ | $F = TD$ |
|-----|-----|-----|-------------------------|--------------------------|-----|-----|----------|
| 1 | 0.0247 | 0.6296 | $5.8 \times 10^{-4}$ | 0.0334 | 4.6665 | 8.3574 | 39.0007 |
| 2 | 0.5053 | 0.1498 | 3.1410 | 179.9666 | 4.4007 | 5.4207 | 23.8555 |
| 3 | 0.4759 | 0.1065 | 1.7217 | 98.6445 | 5.0208 | 3.7432 | 18.7944 |
| 4 | 0.1766 | 0.4928 | 3.1410 | 179.9666 | 3.4664 | 5.3417 | 18.5169 |
| 5 | 0.0955 | 0.4045 | 2.2916 | 131.3014 | 4.9353 | 3.5744 | 17.6409 |
| 6 | 0.5618 | 0.1019 | 1.7217 | 98.6445 | 4.8159 | 3.4912 | 16.8138 |
| 7 | 0.0763 | 0.4486 | 2.8281 | 162.0373 | 4.7391 | 3.4036 | 16.1303 |
| 8 | 0.2616 | 0.2518 | 0.8500 | 48.6986 | 4.9647 | 3.2298 | 16.0356 |
| 9 | 0.4894 | 0.2286 | 3.1410 | 179.9666 | 3.4134 | 4.2434 | 14.4847 |

**Table 2.** Local maxima of the function $F = T + D$.

| No. | $r$ | $d$ | $\beta_{\text{radian}}$ | $\beta_{\text{degrees}}$ | $T$ | $D$ | $F = T + D$ |
|-----|-----|-----|------------------------|-------------------------|-----|-----|-------------|
| 1 | 0.0264 | 0.6286 | $5.83 \times 10^{-4}$ | 0.0334 | 4.5927 | 8.4594 | 13.0521 |
| 2 | 0.5052 | 0.1526 | 3.1410 | 179.9666 | 4.3621 | 5.4639 | 9.8260 |
| 3 | 0.1737 | 0.5020 | 3.1410 | 179.9666 | 3.4109 | 5.4120 | 8.8229 |
| 4 | 0.4765 | 0.1054 | 1.7217 | 98.6445 | 5.0289 | 3.7363 | 8.7652 |
| 5 | 0.0956 | 0.4028 | 2.2916 | 131.3014 | 4.9470 | 3.5644 | 8.5114 |
| 6 | 0.1326 | 0.4436 | 2.7163 | 155.6340 | 4.0975 | 4.1075 | 8.2050 |
| 7 | 0.2620 | 0.2471 | 0.8500 | 48.6986 | 4.9940 | 3.2060 | 8.1999 |
| 8 | 0.0607 | 0.4515 | 2.2916 | 131.3014 | 4.9077 | 3.2713 | 8.1790 |

**Table 3.** Local maxima of the function $F = T + 10D$.

| No. | $r$ | $d$ | $\beta_{\text{radian}}$ | $\beta_{\text{degrees}}$ | $T$ | $D$ | $F = T + 10D$ |
|-----|-----|-----|------------------------|-------------------------|-----|-----|---------------|
| 1 | 0.0290 | 0.6271 | $5.83 \times 10^{-4}$ | 0.0334 | 4.4824 | 8.5175 | 89.6577 |
| 2 | 0.5042 | 0.1642 | 3.1410 | 179.9666 | 4.1982 | 5.5524 | 59.7217 |
| 3 | 0.1689 | 0.5192 | 3.1410 | 179.9666 | 3.2977 | 11.4688 | 58.0176 |
| 4 | 0.4065 | 0.3040 | 2.3587 | 135.1434 | 3.1912 | 4.2054 | 45.2453 |

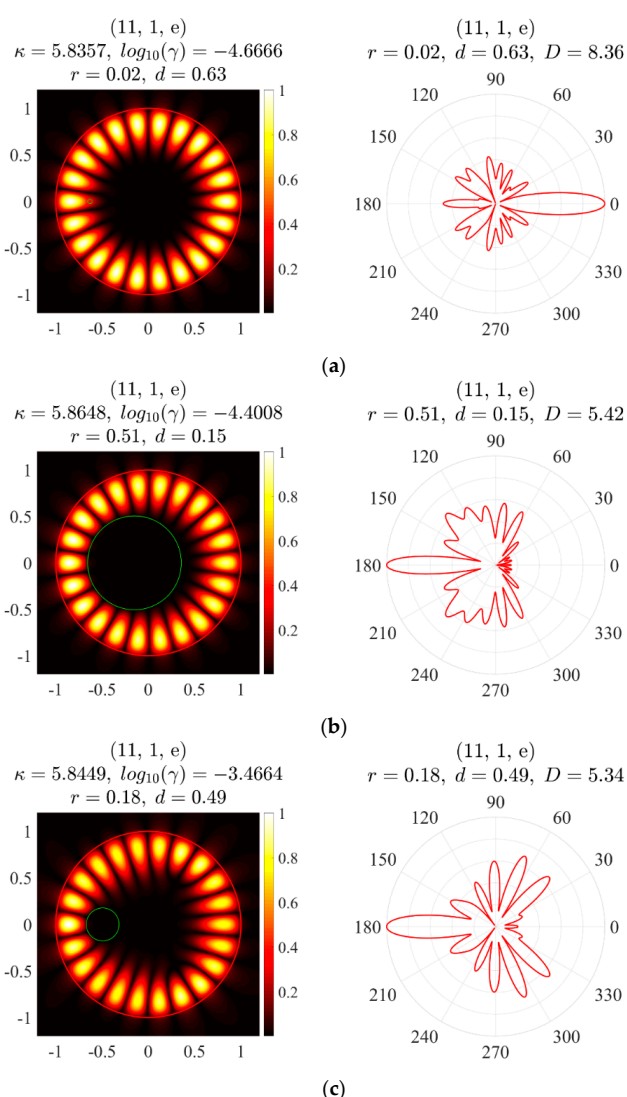

**Figure 6.** Near-field patterns (**left panels**) and far-field patterns (**right panels**) for even mode (11, 1), (**a**) $r = 0.02, d = 0.63$, (**b**) $r = 0.51, d = 0.15$, (**c**) $r = 0.18, d = 0.49$.

Figure 6 shows the near- and far-field patterns for the points selected by intersecting the sets of the local maxima of the considered target functions (25)–(27), among which the pair 0.02 and 0.63 is of the greatest practical interest, because the directivity *D* is maximal, and the value of the threshold is the smallest among all three points. As it is known [20], in the case where the center of the cavity and the center of the hole coincide, the directivity factor is *D* = 2.00 and *T* = 5.2074. This means that if choosing any of the considered pairs of values (*r*,*d*), the directivity *D* becomes at least 2.5 times higher, while the threshold $\gamma$ does not increase significantly.

Thus, in the course of the numerical experiments, we found that a quasi-unidirectional emission could occur both at a small hole radius and at a relatively large hole radius. In the case of a small hole radius, the main beam was directed oppositely to the direction of the hole shift, while in the case of a large hole, the main beam was in the same direction as the hole shift. The maximum directivity was obtained with a small relative radius of the piercing hole. These phenomena were studied by physical experiments in [22].

## 5. Conclusions

We presented the main steps in the reduction of GCFEP for a 2D laser with a piercing hole to a set of four coupled boundary integral equations of the Muller type. We explained the discretization of these equations with the Galerkin method and proved its convergence. We obtained the error estimates for the approximate eigenvalues and eigenfunctions dependent on the smoothness of the generalized eigenfunctions.

Finally, we calculated the on-threshold characteristics of the lasing modes of a circular microcavity with a shifted hole. In the numerical experiments, we varied the position of the piercing hole and the radius of the hole, and computed the changes in the lasing frequencies, directionalities, and thresholds. Our numerical investigation showed that a hole with a suitable radius located at a certain place could lead to notable growth of the directivity of the perturbed whispering-gallery mode emission, together with the preservation of its low threshold. Hence, a piercing hole radius and position in the 2D eccentric microcavity laser can be used as an engineering tool to efficiently control the directivity of emission.

**Author Contributions:** Conceptualization, E.M.K.; methodology, E.M.K., and S.I.S.; software, A.O.S.; validation, I.V.K.; formal analysis, A.I.R.; investigation, I.V.K., A.I.R., and A.O.S.; writing—original draft preparation, A.O.S., A.I.R., I.V.K., and S.I.S.; writing—review and editing, E.M.K.; visualization, I.V.K. and A.O.S.; supervision, E.M.K.; project administration, E.M.K.; funding acquisition, A.O.S. and S.I.S. All authors have read and agreed to the published version of the manuscript.

**Funding:** This work was supported by the Russian Foundation for Basic Research, project nos. 18-41-160029 and 20-08-01154 (Sergey I. Solov'ev), and the Kazan Federal University Strategic Academic Leadership Program.

**Acknowledgments:** The authors thank Alexander I. Nosich, from the Laboratory of Micro and Nano Optics, Institute of Radio-Physics and Electronics, NASU, for proposing the form of the target functions (25)–(27).

**Conflicts of Interest:** The authors declare no conflict of interest.

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
