# Peer review of "Exponentially Convergent Galerkin Method for Numerical Modeling of Lasing in Microcavities with Piercing Holes"

_axioms, doi:10.3390/axioms10030184_

Round 1

Reviewer 1 Report

This paper continues and improves a series of results obtained by the authors in [19-21]. They reduced the   Generalized Complex-Frequency Eigenvalue Problem for a 2-D laser with a  piercing hole to a nonlinear eigenvalue problem for a system of four Muller boundary integral equations (see (16)).       The goal of this article is to present a proof,  based on the fundamental results of the theory of  holomorphic operator-valued functions, of convergence of the trigonometric Galerkin method proposed previously and to estimate the approximate eigenvalues and eigenfunctions.  

The paper is interesting and well-written.  The proofs are clear and easy to follow.  Interesting numerical investigation is given.

As two minor remarks,  I proposed to the authors to add, in p. 6, after (16), the exact conditions imposed  on the operator A and in p. 3, line 2 to replace “section 3” with “Section 3”.

Reviewer 2 Report

This paper proposes the steps to reduce GCFEP for a 2-D laser with a piercing hole to a nonlinear eigenvalue problem for a system of Muller boundary integral equations. Overall, the paper is well written and presented in a logical manner. Some minor typos are listed below.

  • (line 116) “…. and if he region …”

Comment: check for typo.

  • (Line 358) “… tance d between …”

Comment: check for typo.

Reviewer 3 Report

The authors investigate an algorithm for numerical solution of a parametric eigenvalue problem for the Helmholtz equation, motivated by some possible applications. They reduce the issue to a nonlinear eigenvalue problem for a system of Muller boundary integral equations. The solution of it is estimated by numerical methods with applications of a trigonometric Galerkin approach. They also demonstrate the practical effectiveness of the algorithm.

The paper is well written, with a good introduction. Maybe some parts of the proofs should be made more detailed, because authors only very briefly recall some results that are applied there. But they are clear and acceptable. 

Generally the paper is publishable after an improvement of English and a minor revision of some statements of theorems. Namely, Theorem 1 (1.) could be formulated as:

"If k_0 is an eigenvalue of A(k), then for each n\in N there exists an eigenvalue k_n of A_n(k) such that k_n\to k_0."

Analogously Theorem 1 (2.)

It should be mentioned what is n in (23). For instance "Let us denote" could be replaced by: "For each n\in N we denote".

The sentence in Theorem 2: "Here, ... is the generalized eigenspace ..." I suggest to insert before Theorem 2.
